# Efficient Elliptic-Curve-Cryptography-Based Anonymous Authentication for Internet of Things: Tailored Protocols for Periodic and Remote Control Traffic Patterns

**DOI:** 10.3390/s25030897

**Published:** 2025-02-02

**Authors:** Shunfang Hu, Yuanyuan Zhang, Yanru Guo, Yanru Chen, Liangyin Chen

**Affiliations:** College of Computer Science, Sichuan University, Chengdu 610065, China; hsf@ymu.edu.cn (S.H.); yuanyuanzhang@stu.scu.edu.cn (Y.Z.); guoyanru@stu.scu.edu.cn (Y.G.); chenyanru@scu.edu.cn (Y.C.)

**Keywords:** anonymity, authentication and key agreement, traffic pattern, periodic update pattern, remote control pattern, desynchronization, internet of things

## Abstract

IoT-based applications require effective anonymous authentication and key agreement (AKA) protocols to secure data and protect user privacy due to open communication channels and sensitive data. While AKA protocols for these applications have been extensively studied, achieving anonymity remains a challenge. AKA schemes using one-time pseudonyms face resynchronization issues after desynchronization attacks, and the high computational overhead of bilinear pairing and public key encryption limits its applicability. Existing schemes also lack essential security features, causing issues such as vulnerability to ephemeral secret leakage attacks and key compromise impersonation. To address these issues, we propose two novel AKA schemes, PUAKA and RCAKA, designed for different IoT traffic patterns. PUAKA improves end device anonymity in the periodic update pattern by updating one-time pseudonyms with authenticated session keys. RCAKA, for the remote control pattern, ensures anonymity while reducing communication and computation costs using shared signatures and temporary random numbers. A key contribution of RCAKA is its ability to resynchronize end devices with incomplete data in the periodic update pattern, supporting continued authentication. Both protocols’ security is proven under the Real-or-Random model. The performance comparison results show that the proposed protocols exceed existing solutions in security features and communication costs while reducing computational overhead by 32% to 50%.

## 1. Introduction

The Internet of Things (IoT) facilitates seamless interaction between smart sensors, actuators, and servers, enabling autonomous operation without human intervention and supporting a wide range of innovative applications [1]. Currently, IoT-based applications are being increasingly deployed in diverse sectors such as industry, agriculture, transportation, environmental protection, healthcare, and security. In these applications, systems typically involve lightweight sensors, actuators, and more powerful servers. Sensor nodes are deployed in designated areas to periodically monitor and collect real-time data on specific events or changes in the environment and transmit them to the server for analysis and processing or to the actuator side for control. Actuators, in turn, receive data or commands that enable the operation of the instruments or devices in which they are embedded. The gateway connects various devices such as sensors and actuators through various wired or wireless means to achieve comprehensive data coverage and efficient transmission. The server is the core of the entire IoT architecture and contains functions such as device management, data visualization, and remote control to facilitate the management and monitoring of IoT devices. However, data exchanged between sensors and actuators in IoT systems typically transmit over potentially unsecured communication channels. The openness of these channels introduces significant security vulnerabilities [2].

Authentication and key agreement (AKA) protocols are essential for secure IoT communications. They enable end devices to authenticate with each other, securely exchange keys, and verify the integrity of messages. These protocols also create shared session keys between end devices and servers to maintain subsequent communications [3]. The increasing adoption of IoT-based applications has raised significant concerns about privacy and security. Effective AKA proposals must be resistant to various attacks, including key compromise impersonation (KCI), replay, impersonation, and ephemeral secret leakage (ESL) attacks [4]. Furthermore, privacy issues have intensified the demand for anonymity in AKA protocols. Anonymity protects the sender’s identity, while untraceability ensures that different communications cannot be linked to the same entity. Both are essential for safeguarding user privacy in IoT environments. Elliptic Curve Cryptography (ECC) has gained popularity in IoT security because of its efficiency. With smaller key sizes, ECC provides robust security while minimizing computational overhead, storage needs, and bandwidth usage—key advantages for resource-constrained IoT devices. Therefore, designing efficient and secure ECC-based anonymous AKA protocols is essential to improve the security and robustness of IoT-based systems.

Recent efforts have aimed to address security concerns in the ECC-based anonymous AKA protocol for IoT-based applications. In 2016, Tsai et al. [5] presented an anonymous AKA scheme with ECC bilinear pairing for smart grids. However, the protocol has high computational overhead for bilinear pairing and lacks message integrity. He et al. [6] subsequently improved the Tsai et al. protocol to reduce computational and communication costs, but it lacks anonymity and remains vulnerable to ESL attacks. In 2018, Odelu et al. [7] demonstrated that the scheme [5] cannot withstand ESL attacks and proposed an alternative based on bilinear pairing. However, the alternative cannot cope with ESL attacks and does not provide message integrity or anonymity. Saeed et al. [8] introduced an AKA scheme for secure IoT communication between end devices and cloud servers, which offers verification of data integrity, but it is still subject to ESL attacks and does not provide anonymity. In 2020, Garg et al. [9] presented an AKA solution for smart grids that uses fully halved Menezes–Qu–Vanstone for participant authentication. However, Chaudhry et al. [10] later noted that this scheme is susceptible to KCI attacks and does not support anonymity or perfect forward secrecy (PFS). Chaudhry et al. [10] then introduced a new protocol that uses ECC and symmetric encryption for smart grids but remains vulnerable to key escrow problems and Man-in-the-Middle attacks. In 2020, Chaudhry et al. [11] introduced the use of one-time pseudonyms to ensure anonymity for smart grids, but this creates challenges in synchronization. If a pseudonym is blocked or lost, the sensor cannot re-authenticate without re-registration. Similar resynchronization issues are present in the schemes of Park et al. [12] and Zhang et al. [13]. In 2023, Hu et al. [14] highlighted vulnerabilities in the scheme [8], such as the non-resistance of ESL attacks and the absence of anonymity. They then presented an AKA protocol for smart meters and virtual server nodes. However, Wu et al. [15] identified that the protocol [14] suffers from KCI attacks and lacks untraceability. Wu et al. [15] then presented an improved scheme, but it suffers from high computational overhead and susceptibility to Denial of Service (DoS) attacks. The anonymity methods used by both Hu et al. [14] and Wu et al. [15] lead to poor usability due to the server’s need to enumerate stored identities for verification. In 2024, Hu et al. [16] introduced an anonymous AKA protocol for the IoT-based system, using temporary public key encryption to improve anonymity. Although this approach strengthens security, it significantly increases computational overhead, which poses challenges for resource-constrained IoT environments. In general, despite extensive research, achieving effective anonymity in IoT systems remains a significant challenge. Protocols using one-time pseudonyms face serious resynchronization issues after desynchronization attacks, while public key encryption and bilinear pairing introduce high computational costs. Many existing protocols also remain vulnerable to ESL and other security threats.

In contrast to traditional human-type communication applications, IoT-based systems exhibit unique traffic patterns [17]. Specifically, IoT communication often follows periodic update patterns and remote control patterns, where sensors send regular status updates or receive commands from the server for remote control. These patterns differ significantly from conventional human-driven communication behaviors, which are typically more dynamic and less predictable. This distinction is critical when designing AKA protocols for IoT-based applications, as traditional approaches are often designed with human communication patterns in mind, leading to inefficiencies when applied to IoT-based systems.

To address these challenges, we propose a pair of secure anonymous AKA protocols, PUAKA and RCAKA, designed specifically for different IoT traffic patterns. PUAKA is optimized for the periodic update pattern, where sensors periodically send data to the server. It uses authenticated session key parameters to update one-time pseudonymous identities, ensuring sensor anonymity with minimal communication overhead. RCAKA, on the other hand, is suited for the remote control pattern, where the server issues commands to control sensors. RCAKA uses shared signatures and temporary random numbers to maintain anonymity while minimizing computational costs. RCAKA can also resynchronize unfinished sensors in the periodic update pattern and perform authentication and session key negotiation.

The contributions of this paper are as follows:(1)We review existing ECC-based anonymous AKA schemes for IoT-based systems and identify key limitations, such as lack of resynchronization mechanisms and high computational overhead.(2)We introduce two novel AKA protocols, PUAKA and RCAKA, tailored to traffic patterns. PUAKA ensures efficient anonymous authentication in the periodic update pattern, while RCAKA supports both anonymous authentication and resynchronization in the remote control pattern.(3)We provide a formal security analysis of our proposals, demonstrating mutual authentication, anonymity, PFS, and resilience against ESL, KCI, and impersonation attacks.(4)Our protocols significantly reduce computational and communication overhead while offering more robust security features compared to existing schemes.

The paper is structured as follows: In Section 2, we describe the network model, the complexity assumptions, and the traffic patterns. The proposed schemes are introduced in Section 3. Section 4, Section 5 and Section 6 offer detailed security analysis and performance comparisons. Finally, Section 7 concludes the paper.

## 2. Preliminaries

### 2.1. Network Model

As shown in Figure 1, IoT-based application systems consist of components such as end devices, gateways, and servers [18]. End devices typically include sensors and actuators. Sensors are deployed in designated areas to monitor and collect real-time data on environmental changes or events at intervals set by the server. Then, these data are transmitted to servers for analysis or control of actuators. Actuators, in turn, receive data or commands to operate embedded instruments or devices. The gateway connects various devices, such as sensors and actuators, through wired or wireless methods to ensure extensive data coverage and efficient transmission. The server acts as the central hub of the IoT architecture, providing device management, data visualization, and remote control to monitor and manage IoT devices. An IoT-based application system may include multiple servers, and trust authorities (TA) act as dedicated servers for key generation, dissemination, and management. The proposed protocols involve $S_{i}\left(1\leq i\leq l\right)$, $SP_{j}\left(1\leq j\leq m\right)$, and $TA_{k}\left(1\leq k\leq n\right)$, where l≫n and m≫n, with *l*, *m*, and *n* representing the number of end devices, servers, and TAs, respectively.

### 2.2. Complexity Assumptions

Given a large prime *p*, an elliptical curve $E(F_{p})$ is chosen. The points in *E*, along with the identity point *O*, form an additive group *G* of order *q*, with *P* as a generator of *G*.

**Definition 1.** 
*Elliptic Curve Discrete Logarithm Problem (ECDLP): Given two points A and P in $E(F_{p})$, the ECDLP is to find the positive integer x such that X=x·P. The probability that an algorithm A can solve the ECDLP within time t is negligible for a sufficiently small ϵ, and is given by*

(1)
$$Adv_{\mathrm{ECDL}}^{A}\left(t\right)=Pr\left[A\left(P,xP\right)=x:x\in Z_{q}^{*}\right]<\epsilon$$



**Definition 2.** 
*Elliptic Curve Computational Diffie–Hellman Problem (ECDHP): Given three points, P,X=x·P, and Y=y·P in $E(F_{p})$, computing xy·P is considered computationally infeasible. The probability that an algorithm A can solve this problem within time t is negligible for a sufficiently small ϵ, and is given by*

(2)
$$Adv_{\mathrm{CDH}}^{A}\left(t\right)=Pr\left[A\left(xP,yP\right)=xyP:x,y\in Z_{q}^{*}\right]<\epsilon$$



### 2.3. IoT Traffic Pattern

Nikaein et al. [17] analyzed the functions of most applications and identified the traffic patterns listed below.

(1)Periodic update: sensors and actuators send updated status reports, such as smart meter readings (e.g., gas, electricity, water), to the server periodically at intervals configured by the server.(2)Remote control: The server sends commands to the sensors and actuators to control them remotely, such as remotely starting or stopping smart home devices.(3)Event driven: The sensor sends real-time emergency messages to the server when a parameter exceeds a threshold and a given phenomenon occurs, such as a fire or tsunami.

In the event-driven pattern, participants must share symmetric keys in advance to ensure timely data transmission as soon as an event occurs. In this paper, we focus on periodic update and remote control patterns.

## 3. Proposed Scheme

This section provides an overview of the proposed schemes, covering initialization, registration, authentication, and key agreement. The processes of PUAKA and RCAKA are described in the authentication phase. By default, RCAKA runs in remote control mode, by setting type=RC. If RCAKA runs in periodic update mode for resynchronization, type=PU is set. Table 1 shows some symbols of the proposed schemes.

### 3.1. Initialization

The TA follows these processes to initialize the system:T1:The TA chooses a curve $E(F_{p})$ with a base point *P*, and the additive group *G* of order *q*.T2:The TA chooses two one-way hash functions:-$h:{\left\{0,1\right\}}^{*}\to {\left\{0,1\right\}}^{l}$, which is used to generate the hash values and the verifier.-$h_{1}:{\left\{0,1\right\}}^{*}\to {\left\{0,1\right\}}^{64}$, which is used to generate the one-time pseudonym identity.T3:Finally, the TA loads the parameters $\left\{\left(E\left(F_{p}\right),P,q,h_{1},h\right)\right\}$, along with their own identifier, onto each server and end device.

### 3.2. Registration

The end device $S_{s}$ and the server $SP_{sp}$ register with the TA as follows:R1:$S_{s}$ selects a random $z_{s}\in Z_{q}^{*}$, computes $Z_{s}=z_{s}\cdot P$, and transmits the registration request $\left\{ID_{s},Z_{s}\right\}$ to the TA through a secure channel. Likewise, $SP_{sp}$ selects $z_{sp}$, computes $Z_{sp}=z_{sp}\cdot P$, and sends $\left\{ID_{sp},Z_{sp}\right\}$ to the TA.R2:The TA chooses a random $z_{s}^{ta}\in Z_{q}^{*}$ for $S_{s}$ with a valid identifier $ID_{s}$ and calculates the public key $Y_{s}=Z_{s}+z_{s}^{ta}\cdot P$ and the one-time pseudonym identity $TC_{ssp}=h_{1}\left(ID_{s}∥z_{s}^{ta}\right)$ for the S. Similarly, the TA selects $z_{sp}^{ta}$ for $SP_{sp}$ and computes $Y_{sp}=Z_{sp}+z_{sp}^{ta}\cdot P$.R3:The TA stores $\left\{ID_{s},Y_{s}\right\}$ and $\left\{ID_{sp},Y_{sp}\right\}$. Then, the TA transmits $\left\{TC_{ssp},Y_{s},z_{s}^{ta},ID_{sp},Y_{sp}\right\}$ to $S_{s}$ and $\left\{Y_{sp},z_{sp}^{ta},ID_{s},TC_{ssp},Y_{s}\right\}$ to $SP_{sp}$ securely.R4:Once the response is received, $S_{s}$ computes $y_{s}=\left(z_{s}+z_{s}^{ta}\right)\;\mathrm{mod}\;\;q$ as its private key and checks if $Y_{s}=y_{s}\cdot P$. If true, the S computes the signature shared with the SP $X_{sp}=y_{s}\cdot Y_{sp}$ and stores $\left\{ID_{s},Y_{s},y_{s},TC_{ssp},ID_{sp},X_{sp}\right\}$.

Similarly, the SP computes $y_{sp}=\left(z_{sp}+z_{sp}^{ta}\right)\;\mathrm{mod}\;\;q$, $Y_{sp}=y_{sp}\cdot P$, and $X_{s}=y_{sp}\cdot Y_{s}$. Then, the SP initializes the pattern of RCAKA, type=RC. Finally, the SP stores $\left\{ID_{sp},Y_{sp},y_{sp},ID_{s},TC_{ssp},type\right\}$. When a blocked or new end device, $S_{s}^{\prime }$, registers, the TA delivers $\left\{ID_{s}^{\prime },TC_{ssp}^{\prime },Y_{s}^{\prime }\right\}$ to $SP_{sp}$ through secure channels.

**Remark 1.** 
*Signatures $X_{sp}$ and $X_{s}$ are equal, where $X_{sp}=y_{s}\cdot Y_{sp}=y_{s}y_{sp}\cdot P=y_{sp}y_{s}\cdot P=X_{s}$.*


### 3.3. Authentication and Key Agreement

The PUAKA and RCAKA authentication and key agreement processes between $S_{S}$ and $SP_{s}p$ are described in this subsection.

#### 3.3.1. PUAKA

PUAKA is designed for the periodic update pattern, as illustrated in Figure 2, using authenticated session key parameters to update one-time pseudonym identities.

PA1:First, $S_{s}$ selects a random nonce $w_{s}\in Z_{q}^{*}$, computes $W_{s}=w_{s}\cdot Y_{s}$, and generates its verifier $V_{s}=h(ID_{s}∥TC_{ssp}∥W_{s}∥X_{sp})$. Finally, $S_{s}$ sends the authentication request $M_{s}^{1}=\left\{W_{s},TC_{ssp},V_{s}\right\}$ to $SP_{sp}$.PA2:In response, $SP_{sp}$ first checks the received $TC_{ssp}$ to identify the sender. If $TC_{ssp}$ is not recognized, the session is terminated. Next, $SP_{sp}$ verifies the completeness of $M_{s}^{1}$ and the effectiveness of $S_{s}$ by confirming that $V_{s}=h(ID_{s}∥TC_{ssp}∥W_{s}∥X_{s})$ is valid. If this verification fails, $SP_{sp}$ aborts.PA3:Then, $SP_{sp}$ selects a random nonce $w_{sp}\in Z_{q}^{*}$, computes $W_{sp}=w_{sp}\cdot Y_{sp}$, and derives the session key $SK_{sp}=\left(w_{sp}\cdot y_{sp}\;\mathrm{mod}\;q\right)\cdot W_{s}$. The session key is then computed as $SSK_{sp}=h(ID_{s}∥ID_{sp}∥SK_{sp})$. $SP_{sp}$ updates the pseudonym identity of *S* as $TC_{ssp}=$
$TC_{ssp}⊕h\left(SK_{sp}∥W_{sp}\right)$ and calculates its verifier $V_{sp}=h(X_{s}∥ID_{sp}∥TC_{ssp}∥W_{sp}∥SK_{sp})$. Finally, $SP_{sp}$ sends the reply message $M_{sp}^{1}=\left\{W_{sp},V_{sp}\right\}$ to $S_{s}$.PA4:Upon receiving the response, $S_{s}$ computes the session key $SK_{s}=\left(\left(w_{s}\cdot y_{s}\;\mathrm{mod}\;q\right)\cdot W_{sp}\right)$ and updates the pseudonym identity as $TC_{ssp}^{\mathrm{new}\;}=TC_{ssp}⊕h\left(SK_{s}∥W_{sp}\right)$. $S_{s}$ then verifies the equivalence $V_{sp}=h(X_{sp}∥ID_{sp}∥TC_{ssp}^{new}∥W_{sp}∥SK_{s})$. If the check is satisfied, $S_{s}$ derives the session key $SSK_{s}=h(ID_{s}∥ID_{sp}∥SK_{s})$, updates $TC_{ssp}$ with $TC_{ssp}^{\mathrm{new}\;}$, and completes the authentication and session key agreement.

#### 3.3.2. RCAKA

RCAKA is designed for the remote control pattern, as depicted in Figure 3. During the execution of PUAKA, if the one-time pseudo-identity synchronization fails or messages are blocked during authentication, scheduled data updates may not be completed within the configured intervals. In such cases, the server can resynchronize the end devices using RCAKA to finalize the authentication and the session key agreement.

RA1:First, $SP_{sp}$ generates a random nonce $w_{sp}\in Z_{q}^{*}$ and a timestamp $T_{sp}$. Next, $SP_{sp}$ computes $W_{sp}=w_{sp}\cdot Y_{sp}$.If the type is PU, $SP_{sp}$ selects another random $ww_{s}\in Z_{q}^{*}$ and constructs a new pseudonym identity for the client *S* as $TC_{ssp}=h_{1}\left(ID_{s}∥ww_{s}\right)$. Subsequently, $SP_{sp}$ masks $TC_{ssp}$ by computing $ETC_{ssp}=TC_{ssp}⊕h\left(X_{s}∥W_{sp}\right)$, where $X_{s}$ is a shared secret between $SP_{sp}$ and $S_{s}$. Furthermore, $SP_{sp}$ computes the verifier $V_{sp}=h(ID_{sp}∥TC_{ssp}∥W_{sp}∥X_{s}∥T_{sp})$. Finally, $SP_{sp}$ sends the resynchronization and authentication request message $M_{sp}^{2}=\left\{W_{sp},ETC_{ssp},V_{sp},T_{sp}\right\}$ to $S_{s}$.If the type is not PU, $SP_{sp}$ calculates a verifier $V_{sp}=h(ID_{sp}∥W_{sp}∥X_{s}∥T_{sp})$ and sends the authentication request message $M_{sp}^{2}=\left\{W_{sp},V_{sp},T_{sp}\right\}$ to $S_{s}$.RA2:When receiving $M_{sp}^{2}$, $S_{s}$ first verifies the freshness of $T_{sp}$. Next, if $M_{sp}^{2}=\left\{W_{sp},ETC_{ssp},V_{sp},T_{sp}\right\}$, $S_{s}$ de-masks $TC_{ssp}^{new}=ETC_{ssp}⊕h\left(X_{sp}∥W_{sp}\right)$, then checks $V_{sp}=h(ID_{sp}∥TC_{ssp}^{new}∥W_{sp}∥X_{sp}∥T_{sp})$. If the condition is satisfied, $S_{s}$ updates $TC_{ssp}$ with $TC_{ssp}^{new}$. Otherwise, $S_{s}$ verifies if $V_{sp}=h(ID_{sp}∥W_{sp}∥X_{sp}∥T_{sp})$. If it is not satisfied, $S_{s}$ will terminate.RA3:$S_{s}$ begins by selecting a random nonce $w_{s}\in Z_{q}^{*}$ and generating a timestamp $T_{s}$. Secondly, $S_{s}$ calculates $W_{s}=w_{s}\cdot Y_{s}$ and derives the session key $SK_{s}=\left(w_{s}\cdot y_{s}\;\mathrm{mod}\;q\right)\cdot W_{sp}$. The session key is used to compute $SSK_{s}=h(ID_{s}∥ID_{sp}∥SK_{s})$. Next, $S_{s}$ masks its identity using $EID_{s}=ID_{s}⊕h\left(W_{s}∥SK_{s}\right)$. Afterward, $S_{s}$ computes its verifier as $V_{s}=h(ID_{s}∥SSK_{s}∥W_{s}∥T_{s})$. Finally, $S_{s}$ sends the reply message $M_{s}^{2}=$
$\left\{W_{s},T_{s},EID_{s},V_{s}\right\}$ to $SP_{sp}$.RA4:Upon receiving $M_{s}^{2}$, $SP_{sp}$ first confirms the freshness of $T_{s}$. Then, $SP_{sp}$ calculates the session key $SK_{sp}=\left(w_{sp}\cdot y_{sp}\;\mathrm{mod}\;q\right)\cdot W_{s}$ and recovers $ID_{s}$ by de-masking $EID_{s}$ using $ID_{s}=$
$EID_{s}⊕h\left(W_{s}∥SK_{sp}\right)$. Next, $SP_{sp}$ computes the session key $SSK_{sp}=h(ID_{s}∥ID_{sp}∥SK_{sp})$. Finally, $SP_{sp}$ verifies the equivalence of the verifier: $V_{s}=h(ID_{s}∥SSK_{sp}∥W_{s}∥T_{s})$. If the condition does not hold, the session is terminated.

## 4. Formal Security Proof

Taking PUAKA as an example, this section discusses its security using the ROR model [19].

### 4.1. Participant

In the PUAKA protocol, there are two main participants: an end device *S* and a server SP. Each participant might involve a number of oracles involved in individual parallel implementations of PUAKA. Each oracle is represented as $S_{i}$ for the end device and $SP_{j}$ for the server, where i,j∈Z. A general oracle is denoted as I∈SP∪S. The messages exchanged by oracle *I* define its session identifier, denoted as Sid.

An oracle can be in one of three potential states:Accept: An oracle *I* reaches state accept when it receives the latest expected protocol message.Reject: If the oracle accepts an incorrect message, it enters the reject state.⊥: If *I* does not receive any response, then it switches to the state ⊥.

### 4.2. Adversary Model

Under the eCK adversary model [20], A controls the public channels. In addition, A can learn the secrets of $S_{i}$ and $SP_{j}$. A interacts with the oracles through the following queries [21]:

(1)$Execute(I_{i})$: A can obtain the messages $\left\{W_{s},ETC_{ssp},V_{s}\right\}$ from $S_{i}$ and $\left\{W_{sp},V_{sp}\right\}$ from $SP_{j}$.(2)$Send(m_{I},I)$: A transmits a message $m_{I}$ to *I* and receives a response according to the PUAKA protocol.(3)Corrupt(I): A has the ability to compromise *I* and retrieve its long-term secrets.(4)SKReveal(I): The session key owned by *I* can be obtained by A.(5)ESReveal(I): The ephemeral secrets of *I* can be acquired by A.(6)h(m): The output of a randomized hash for a given message *m* can be obtained for A.(7)Test(I): This query is designed to define the semantic security of the session key. If no session key of *I* is defined, the query returns ⊥. Otherwise, a private coin *d* is flipped. If d=1, the actual session key is returned to A; otherwise, a random value of the same size is returned. The objective of the adversary is to distinguish between the real session key and a random one.(8)Expire(I): The session key held by *I* will be deleted by this query.

Fresh: A session $s_{i}$ is considered fresh if neither the session itself nor its associated partner session has been revealed. If the adversary A issues any of the following queries before invoking Expire(I), the session se is considered exposed, even if it has not yet been fully established: SKReveal(I), ESReveal(I), or Corrupt(I). Once a session is exposed, it is regarded as locally exposed.

*Partner: *$S_{i}$ and $SP_{j}$ will be considered partners if they meet the following conditions: both must reach the accept state, both oracles must be fresh, and they must share the same session identifier Sid and mutually authenticate and agree on the session key.

**Definition 3.** 
*Under the adversarial model of eCK, an AKA scheme is considered semantically secure if Adv(A)≤ϵ for a sufficiently small ϵ.*


### 4.3. Formal Security Analysis

**Theorem 1.** 
*Assuming that the semantic security of PUAKA is to be broken, A can execute a maximum of $q_{s}$
Send() queries, $q_{e}$
Execute() queries, and $q_{h}$
h() queries in time t. In light of a hash length of l, A has the advantage that*

(3)
$$\begin{array}{cc} Adv(A) & \leq \frac{\left(q_{h}^{2}+2q_{s}\right)}{2^{l}}+\frac{{\left(q_{s}+q_{e}\right)}^{2}}{2(q-1)}\\ \; & +\left(\frac{3q_{s}}{q-1}+\frac{3q_{s}}{2^{l}}\right)\max\left\{{Adv}_{\mathrm{ECDL}}^{A}\left(t\right),{Adv}_{\mathrm{ECD}}^{A}\left(t\right)\right\} \end{array}$$



**Proof.** For the proof below, a set of six games $GM_{i}\left(i=0,1,\ldots ,5\right)$ is defined. When A successfully predicts the bit *d* returned by the Test(I) query, event $S_{i}$ appears in each corresponding game.$GM_{0}$: The ROR model simulates an actual attack in this game. Therefore, the $A^{\prime }$ advantage is given by(4)$$Adv\left(A\right)=|2Pr\left[S_{0}\right]-1|$$$GM_{1}$: A can retrieve all messages via the Execute() query, including $\left\{W_{s},TC_{ssp},V_{s}\right\}$ and $\left\{W_{sp},V_{sp}\right\}$. Afterward, A is able to validate the validity of the computed session keys $SSK_{s}$ and $SSK_{sp}$ with the SKReveal(I) and Test(I) queries. $\left\{W_{s},TC_{ssp},V_{s}\right\}$ and $\left\{W_{sp},V_{sp}\right\}$ do not allow inferring with the session key. Therefore, it is infeasible to distinguish between the actual attack and the game $GM_{1}$. Thus,(5)$$Pr\left[S_{1}\right]=Pr\left[S_{0}\right]$$Additionally, three lists are used to track the relevant outcomes:$L_{h}$: This list contains the tuples of <input, output> for all h(·) queries.$L_{h}^{A}$: This list stores the responses to h(·) queries issued by the adversary A.$L_{E}$: This list records the tuples of <input, output> for all Execute (·) queries.${GM}_{2}$: This game models the scenario where A is capable of forging messages that could be accepted via Send (·) and h(·) queries. The semantic security of PUAKA is only threatened when A has detected collisions and successfully sends a valid message. According to the birthday paradox, the collision probability in the h(·) output is bounded by $\frac{q_{h}^{2}}{2^{2+1}}$. The collision probability of random numbers is bounded by $\frac{{\left(q_{s}+q_{e}\right)}^{2}}{2(q-1)}$. Consequently, unless such collisions occur, it is impossible to distinguish between $GM_{2}$ and $GM_{1}$. Therefore,(6)$$|Pr\left[S_{2}\right]-Pr\left[S_{1}\right]|\leq \frac{{\left(q_{s}+q_{e}\right)}^{2}}{2(q-1)}+\frac{q_{h}^{2}}{2^{l+1}}$$$GM_{3}$: This game simulates the scenario where A manages to deduce certain parameters and forge messages $\left\{W_{s},TC_{ssp},V_{s}\right\}$ and $\left\{W_{sp},V_{sp}\right\}$ without using random oracles. Additional operations are introduced in the Send(·) queries for $GM_{3}$. If there is a validation failure, the queries will terminate.
(1)For $send(W_{s},TC_{ssp},V_{s})$, if $(ID_{s}∥TC_{ssp}∥W_{s}∥*,V_{s})\in L_{h}^{A}$, the probability is at most $\frac{q_{\kappa }}{2^{l}}$.(2)For $send(W_{sp},V_{sp})$, if $(*∥ID_{sp}∥*∥W_{sp}∥SK_{sp},V_{sp})\in L_{h}^{A}$, the probability is at most $\frac{q_{\kappa }}{2^{l}}$.If these checks are considered, $GM_{3}$ and $GM_{2}$ are indistinguishable. Thus,(7)$$|Pr\left[S_{3}\right]-Pr\left[S_{2}\right]|\leq \frac{2q_{s}}{2^{l}}$$$GM_{4}$: This game simulates the corruption capacity of A. A cannot determine $SSK_{s}$ or $SSK_{sp}$ unless it captures $\left(y_{s},w_{s}\right)$ or $\left(y_{sp},w_{sp}\right)$. There are four possible cases where A can use execute (·) and h(·) queries to compute the session keys:
(1)A obtains both long-term private keys, $y_{s}$ and $y_{sp}$, by issuing $\mathrm{Corrupt}(S_{i})$ and $\mathrm{Corrupt}(SP_{j})$ queries. Then, A attempts to obtain information about $w_{s}$ and $w_{sp}$ via Send(·) queries. The attack probability is at most $\frac{2q_{s}}{q-1}$.(2)A issues $\mathrm{Corrupt}(S_{i})$ and $\mathrm{ESReveal}(SP_{j})$, then obtains $y_{s}$ and $w_{sp}$. Afterward, A attempts to retrieve information about $w_{s}$ and $y_{sp}$ via Send(·) queries. The attack probability is at most $\frac{q_{s}}{2^{t}}+\frac{q_{*}}{q-1}$.(3)A issues $\mathrm{ESReveal}(S_{i})$ and $\mathrm{Corrupt}(SP_{j})$, then obtains $w_{s}$ and $y_{sp}$. It then attempts to retrieve information about $y_{s}$ and $w_{sp}$ via Send(·) queries. The attack probability is at most $\frac{q_{s}}{q-1}+\frac{q_{s}}{2}$.(4)A issues $\mathrm{ESReveal}(SM_{i})$ and $\mathrm{ESReveal}(SP_{j})$, then obtains $w_{s}$ and $w_{sp}$. A then attempts to retrieve information about $y_{s}$ and $y_{sp}$ via Send(·) queries. The attack probability is at most $\frac{2q_{*}}{2^{t}}$.In all cases, A cannot compute $SSK_{s}$ or $SSK_{sp}$ except if it resolves the ECDL or ECD problems in time *t*. Thus,(8)$$|Pr\left[S_{4}\right]-Pr\left[S_{3}\right]|\leq \left(\frac{3q_{s}}{q-1}+\frac{3q_{s}}{2^{l}}\right)\max\left\{{Adv}_{\mathrm{ECDL}}^{A}\left(t\right),{Adv}_{\mathrm{ECD}}^{A}\left(t\right)\right\}$$$GM_{5}$: The simulation of $GM_{5}$ is identical to that of $GM_{4}$, with the exception that the Test(·) query halts if A makes an $h(ID_{s}|ID_{sp}|SSK_{s})$ query. The maximum probability that A successfully obtains $SSK_{s}$ is bounded by $\frac{q_{h}^{2}}{2^{2}}$. Therefore,(9)$$|Pr\left[S_{5}\right]-Pr\left[S_{4}\right]|\leq \frac{q_{h}^{2}}{2^{2}}$$Unless A provides the correct input for the h(·) query, it will not be able to distinguish between the actual session key and the randomized session key. Hence, the probability is(10)$$Pr\left[S_{5}\right]=\frac{1}{2}$$Considering all probabilities together, we can conclude that Theorem 1 holds. □

## 5. Descriptive Security Analysis

### 5.1. Anonymity

In the PUAKA protocol, the one-time pseudo-identity is updated using the temporary public key $TC_{ssp}^{new}=TC_{ssp}⊕h\left(SK_{s}∥W_{sp}\right)$. This guarantees the anonymity and untraceability of the end device. Similarly, in the RCAKA protocol, the identity is masked with the authenticated session key $EID_{s}=ID_{s}⊕h\left(w_{s}∥SK_{s}\right)$, ensuring the same level of anonymity and untraceability.

### 5.2. Perfect Forward Secrecy

A protocol prevents an adversary from accessing the keys of the previous session, thus ensuring perfect forward secrecy, even if long-term secrets are later compromised. The session key $SSK_{s}=h(ID_{s}∥ID_{sp}∥SK_{s})$, where $SK_{s}=\left(\left(x_{s}y_{s}\right)\;\mathrm{mod}\;q\right)\cdot W_{sp}$ is derived from the long-term secret values and ephemeral parameters. Even if an adversary A gains access to the long-term secrets $y_{s,}\;y_{sp},X_{s}$, and $X_{sp}$, it cannot compute the session key $SSK_{s}$. This is because A does not have access to the ephemeral values $w_{s}$ and $w_{sp}$ required for the derivation of the session key.

### 5.3. Ephemeral Secret Leakage Attack Resistance

As shown in the above subsection, $SSK_{s}=h(ID_{s}∥ID_{sp}∥SK_{s})$, where $SK_{s}=\left(\left(x_{s}y_{s}\right)\;\mathrm{mod}\;q\right)\cdot W_{sp}$. If A gains access to ephemeral secrets $w_{s}$ and $w_{sp}$, it cannot yet determine $SSK_{s}$ because it lacks long-term keys $k_{s}$ and $y_{sp}$.

### 5.4. No Key Escrow Problem

After registration, the end device *S* receives its long-term private key $y_{s}=\left(Z_{s}+z_{s}^{ta}\right)$
modq. Only the TA generates the partial key $z_{s}^{ta}$, ensuring that there is no key escrow problem. The process for the SP is analogous.

### 5.5. IoT Node Capture Attack Resistance

In the event of a capture of an IoT node, A can extract long-term credentials such as $\left\{ID_{s},Y_{s},y_{s},TC_{ssp},ID_{sp},W_{sp}\right\}$. However, since each device has unique credentials, the session key between the compromised device and the associated server $S_{sp}$ is the only part at risk. The security of other session keys remains intact between uncompromised devices and servers.

### 5.6. Key Compromise Impersonation Attack Resistance

Although A has acquired long-term secrets for the end device, it is not possible for it to impersonate the server to perform authentication with the end device. The reason for this is that the shared secret for *S* is calculated as $SSK_{s}=h(ID_{s}∥ID_{sp}∥SK_{s})$, where $SK_{s}=\left(\left(w_{s}\cdot y_{s}\right)\;\mathrm{mod}\;q\right)\cdot W_{sp}$ and $W_{sp}=w_{sp}\cdot Y_{sp}$. The values $w_{s}$ and $w_{sp}$ are random values generated for the session and are critical for calculating the session key. Even with the credentials of *S*, A cannot reconstruct these random values. Without them, A cannot compute the correct session key $SK_{s}$, and therefore cannot complete the authentication process with *S* or impersonate SP.

### 5.7. Impersonation Attack Resistance

In the case of *S*-impersonation attacks, the adversary lacks long-term secrets, including $y_{s},y_{sp},TC_{ssp}$, and $X_{sp}$. As a result, it cannot generate the authentication request message $\left\{W_{s},TC_{ssp},V_{s}\right\}$, which contains $X_{sp}$. Without this information, the adversary cannot impersonate *S*. Similarly, the adversary is also unable to impersonate SP due to the unavailability of the required secrets.

### 5.8. Man-in-the-Middle Attack Resistance

During authentication, the *SP* authenticates the *S* by checking whether $V_{s}=h(ID_{s}∥TC_{ssp}∥W_{s}∥X_{s})$, and the *S* authenticates the *SP* by checking the equivalence of $V_{sp}=h(X_{sp}∥ID_{sp}∥TC_{ssp}^{new}∥W_{sp}∥SK_{s})$, where $WT_{s}=WT_{sp}$ are the shared signatures between the *S* and *SP*, which are only known to the *SP* and *S*. In other words, the scheme is resistant to Man-in-the-Middle attacks.

## 6. Performance Comparison

This section provides a comparison of the computational and communication costs, as well as the security and functional attributes, of the proposed protocols with those of other existing schemes, including [10,14,15,16].

### 6.1. Computation Cost

We assume that $T_{h}$, $T_{pa}$, and $T_{pm}$ denote the runtime of hash computation, point addition, and point multiplication, respectively. The tests were carried out on a server equipped with an Intel Core i5 2.0 GHz processor and 16 GB of RAM running macOS 13.4.1. When using the Curve25519 elliptic curve with a point length of 384 bits and a prime modulus $p=2^{192}$, the average runtime for each operation was as follows: 0.005 ms for hashing, 0.006 ms for point addition, and 1.258 ms for point multiplication. The end device nodes used a Raspberry Pi 3 Model B+ board with an ARM Cortex-A53 1.4 GHz processor and 1 GB of RAM. Under the same test conditions, the measured average runtime was 0.019 ms for hashing, 0.025 ms for point addition, and 2.225 ms for point multiplication.

As illustrated in Table 2, the PUAKA protocol demonstrates the lowest computational overhead for both authentication and key agreement operations. Compared with existing related works, the proposed schemes achieve a significant reduction in computational overhead, with reductions ranging from 32% to 50%. This considerable improvement in efficiency underscores the effectiveness of the proposed approach in optimizing performance, especially in resource-constrained environments where minimizing computational costs is essential to ensure scalability and responsiveness in IoT systems.

### 6.2. Communication Cost

Let *G*, *H*, ID, *R*, and TS denote the lengths of the ECC point, hash output, identity, random number, and timestamp, respectively. Based on the length of communication overheads in studies [14,15,16], these lengths are 384 bits, 128 bits, 64 bits, 128 bits, and 32 bits, respectively. According to Table 3, the proposed solutions significantly reduce communication costs by optimizing message exchange patterns and minimizing the volume of transmitted data. As a result, the proposed schemes achieve the lowest communication overhead when compared to related studies, further improving their suitability for deployment in resource-constrained IoT environments, where reducing communication burden is crucial to improve overall system performance and ensuring faster authentication.

### 6.3. Performance Comparison

As demonstrated in Table 4, the proposed protocol stands out as more prominent compared to the related protocols [10,14,15,16]. The proposed schemes not only offer superior security features, but also provide a range of additional functional attributes that make them more adaptable and efficient in real-world IoT applications.

## 7. Conclusions

We reviewed related ECC-based anonymous AKA schemes for IoT and identified several key limitations. First, anonymous authentication using one-time pseudo-identities suffers from desynchronization attacks because of the absence of resynchronization mechanisms. Second, schemes based on public key cryptography, while offering anonymity, often result in increased computational overhead. In addition, existing solutions fail to address critical security issues, such as vulnerability to ESL and KCI attacks. To address these issues, we propose two secure ECC-based anonymous AKA protocols tailored to IoT traffic patterns: PUAKA and RCAKA. PUAKA operates using a periodic update pattern and leverages one-time anonymous identities to maintain end device anonymity. These identities are refreshed with authenticated session key parameters, reducing communication overhead compared to existing schemes. In contrast, RCAKA follows a remote control pattern, employing shared signatures and temporary random numbers to ensure anonymity while minimizing both communication and computational costs. RCAKA also includes a resynchronization mechanism to update sensors that have not yet completed their data updates, allowing secure authentication and key agreement during the session. Both protocols have been formally analyzed to ensure anonymity and PFS and are secure against attacks such as impersonation, KCI, and ESL. Performance comparison results indicate that the proposed schemes excel in security features and communication costs, reducing computational overhead by 32% to 50% compared to existing schemes.

ECC-based authentication schemes have been regarded as classical security protocols, with increasing concerns about their efficiency, particularly in light of emerging side-channel and quantum-based attacks. Although ECC continues to be widely used in IoT infrastructures, it faces vulnerabilities to these advanced attacks, such as those demonstrated by the Downfall and Inception side-channel attacks, as well as potential risks from quantum computing. Given these concerns, the adoption of recent quantum resistant cryptographic protocols, such as CRYSTALS-Kyber for general encryption and CRYSTALS-Dilithium, FALCON, and SPHINCS+ for digital signatures, represents an important direction for the future of secure authentication schemes. Although our study focuses primarily on ECC-based solutions due to their current efficiency in resource-constrained IoT environments, we recognize the need to incorporate quantum-resistant protocols as part of ongoing and future research efforts to address emerging security threats.

## Figures and Tables

**Figure 1 sensors-25-00897-f001:**
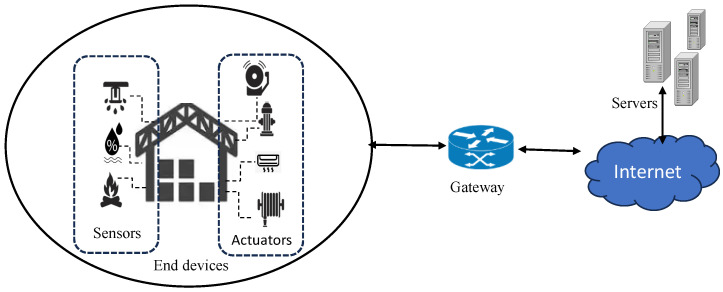
Network model.

**Figure 2 sensors-25-00897-f002:**
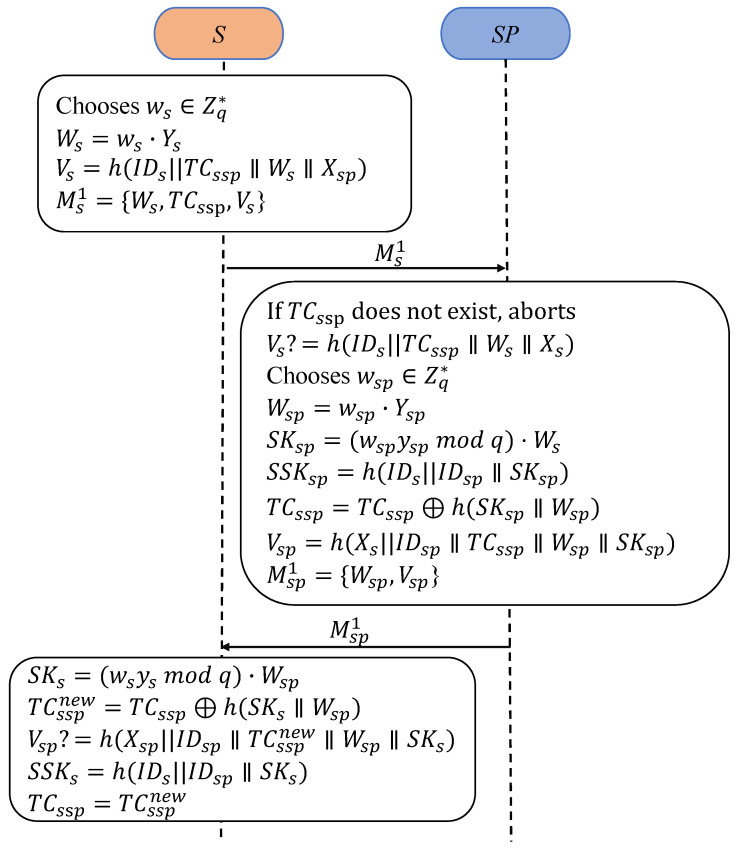
Processes of PUAKA.

**Figure 3 sensors-25-00897-f003:**
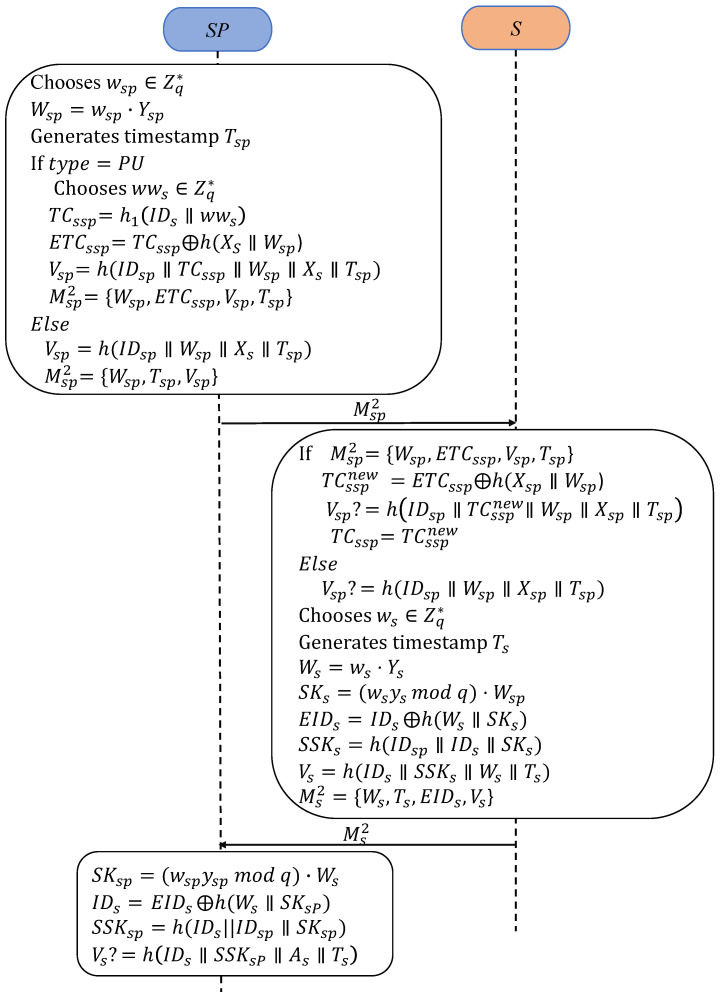
Processes of RCAKA.

**Table 1 sensors-25-00897-t001:** Notations.

Notation	Description	Notation	Description
*S*	Sensor or actuator	SP	Server
TA	Trust authority	A	Adversary
$z_{i}$	Entity-generated partial private key	$z_{i}^{ta}$	TA-generated partial private key
$w_{i}$	Temporary secret of entity *i*	$W_{i}$	Temporary public key of entity *i*
$y_{i}$/$Y_{i}$	Private/public key of entity *i*	$TS_{i}$	Timestamp of entity *i*
$SSK_{i}$	Session key of entity *i*	$TC_{ssp}$	One-time pseudonym identity

**Table 2 sensors-25-00897-t002:** Computation cost.

Scheme	End Device (ms)	Server (ms)	Total (ms)	Decline
PUAKA	2$T_{pm}$ + 4$T_{h}$ ≈7.624	2$T_{pm}$ + 4$T_{h}$ ≈2.536	10.16	-
RCAKA(PU)	2$T_{pm}$ + 5$T_{h}$ ≈7.656	2$T_{pm}$ + 6$T_{h}$ ≈2.546	10.202	-
RCAKA(RC)	2$T_{pm}$ + 4$T_{h}$ ≈7.624	2$T_{pm}$ + 4$T_{h}$ ≈2.536	10.16	-
[10]	4$T_{pm}$ + 4$T_{h}$ ≈15.12	4$T_{pm}$ + 4$T_{h}$ ≈5.052	20.172	49%
[14]	3$T_{pm}$ + $T_{pa}$ + 4$T_{h}$ ≈11.413	3$T_{pm}$ + $T_{pa}$ + 4$T_{h}$ ≈3.8	15.213	33%
[15]	4$T_{pm}$ + 2$T_{pa}$ + 6$T_{h}$ ≈15.266	4$T_{pm}$ + 2$T_{pa}$ + 6$T_{h}$ ≈5.074	20.34	50%
[16]	3$T_{pm}$ + 3$T_{h}$ ≈11.34	3$T_{pm}$ + 3$T_{h}$ ≈3.789	15.129	32%

**Table 3 sensors-25-00897-t003:** Communication cost.

Scheme	End Device (Bits)	Server (Bits)	Total (Bits)
PUAKA	G + H + ID = 608	G + H = 544	1152
RCAKA(PU)	G + H + ID + TS = 640	G + H + ID + TS = 640	1280
RCAKA(RC)	G + H + TS = 576	G + H + ID + TS = 640	1216
[10]	G + H + TS + ID = 640	G + H + TS = 576	1216
[14]	G + H + R + TS + ID = 800	G + H + R + TS + ID = 800	1600
[15]	G + 2H + 2TS + ID = 832	G + H + TS + ID = 640	1472
[16]	G + H + TS + ID = 640	G + H + TS = 576	1216

**Table 4 sensors-25-00897-t004:** Performance comparison.

Scheme	F1	F2	F3	F4	F5	F6	F7	F8	F9	F10	F11	F12	F13	F14
[10]	✔	✔	✔	✔	×	✔	✔	✔	✔	✔	×	✔	✔	×
[14]	×	✔	✔	×	✔	✔	✔	✔	✔	✔	✔	✔	×	×
[15]	✔	✔	✔	×	✔	✔	×	×	✔	✔	✔	✔	✔	×
[16]	✔	✔	✔	✔	✔	✔	✔	✔	✔	✔	✔	✔	✔	×
Ours	✔	✔	✔	✔	✔	✔	✔	✔	✔	✔	✔	✔	✔	✔

F1: KCI attack resistance; F2: IoT node capture attack resistance; F3: impersonation attack resistance; F4: availability; F5: MIM attack resistance; F6: replay attack resistance; F7: desynchronization attack resistance; F8: DoS attack resistance; F9: mutual authentication without RC/TA; F10: PFS; F11: no key escrow problem; F12: ESL resistance; F13: anonymity; F14: high computation cost. ✔: supports functional features or security; ×: does not support functional features or the program is insecure.

## Data Availability

The raw data supporting the conclusions of this article will be made available by the authors on request.

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
