# Peer review of "Efficient Elliptic-Curve-Cryptography-Based Anonymous Authentication for Internet of Things: Tailored Protocols for Periodic and Remote Control Traffic Patterns"

_sensors, 2025, doi:10.3390/s25030897_

Round 1

Reviewer 1 Report

Comments and Suggestions for Authors

This paper propose ECC-based PUAKA and RCAKA schemes, which are suitable for different traffic patterns. The followings are the comments:

1. In section 4.3, the formal security analysis is not written seriously enough, i.e., in page 9, the heading numbers are in disorder.

2. In comparing the communication cost, the authors only consider one version of the lengths of G, H, ID, R, TS. More versions should be added, or the reason for choosing this length version should be stated.

3. In section 2.2, the advantage should be defined to be related to the security parameter n, while this paper only uses epsilon. 

Author Response

Dear Editor and Reviewers,
We are very grateful for your constructive comments on our submission, sensors-3425811.
We have carefully considered all of your comments while preparing the current revision and this response letter.
Below, we provide detailed responses to each of your comments. For clarity, we cite the specific comments and their indices, followed by our responses and the corresponding actions we have taken. In the revised manuscript, we have marked the revised sections with \textcolor{blue}{blue} color and the responded content with \textcolor{red}{red} color.

We hope that these detailed responses will help the editor and reviewers better understand the revisions made to the manuscript. Thank you once again for your thoughtful consideration. We look forward to your feedback.

Sincerely yours,

Shunfang  Hu

Reviewer 2 Report

Comments and Suggestions for Authors

Dear Authors,

Thanks for your contributions to the Authentication area.

After reviewing all the sections of the current version, I have decided to recommend a major revision since there are several writing errors and technical shortages that must be significantly improved before the final decision is made. Below, my comments are summarized, suggesting some areas for improvement.

1) In the literature, the ECC-based authentication schemes are considered classical security protocols and are no longer considered efficient, which is opposed to the claim of the authors. Although they are still in use by the Internet of Everything (IoE) infrastructures, they are prone to being compromised by side-channel (e.g., Downfall and Inception) and quantum-based attacks. Please try to apply recent standard quantum-resistance cryptographic protocols (e.g., CRYSTALS-Kyber for general encryption and CRYSTALS-Dilithium, FALCON, and SPHINCS+ for digital signatures) instead of ECC-based approaches while developing novel authentication schemes.

2) There are abbreviations/acronyms that you did not define their long forms in the first used sections, which confuse readers to understand the concepts. Please proofread all the sections by making sure the abbreviations/accronyms are defined correctly. Since there are many abbreviations/acronyms in this article, it is better to include a table of abbreviations that list the most used abbreviations and their definitions to increase the article's clarity.

3) On page 2, what is a MIM attack? Is it the Man-in-the-Middle attack (MitM)? Why didn't you consider this attack during the attack resistance analysis?

4) In this study, the authors did not consider the side-channel attacks that are one of the effective ways to break  ECC-based protocols, which raises the question about the efficiency of the suggested protocol.

5) In the literature, there are many articles that exactly have the term "Anonymous Authentication for IoT" in their titles, and you did not include any of them in your comparative analysis. This point also raises the question about the quality of your contributions. Please check the aformentioned references.

Comments on the Quality of English Language

Overall, the paper is well-written and organized. However, the following areas need improvement: careful proofreading by the authors.

6) It seems there is no coherency between the paragraphs of the introduction. In a high-level technical paper, you must summarize the problem statement by giving some provable references to why the area needs investigation and highlighting the facts. Then, you must summarize the chosen state-of-the-art schemes, which are close to the core topic of your study, and explain them in a related work section. In my opinion, the current merged introduction and related works sections do not provide enough evidence to enlighten why this area requires further improvements.

7) Please split the long sentences, particularly the ones with more than three lines (e.g., 411-414), into two short sentences and connect them by applying conjunctions.

Author Response

(The authors gave the same response as above.)

Reviewer 3 Report

Comments and Suggestions for Authors

See Review Comment

Comments on the Quality of English Language

no

Author Response

Dear Editor and Reviewers,We are very grateful for your constructive comments on our submission, sensors-3425811.
We have carefully considered all of your comments while preparing the current revision and this response letter.
Below, we provide detailed responses to each of your comments. For clarity, we cite the specific comments and their indices, followed by our responses and the corresponding actions we have taken. In the revised manuscript, we have marked the revised sections with \textcolor{blue}{blue} color and the responded content with \textcolor{red}{red} color.

We hope that these detailed responses will help the editor and reviewers better understand the revisions made to the manuscript. Thank you once again for your thoughtful consideration. We look forward to your feedback.

Sincerely yours,

Shunfang  Hu

Round 2

Reviewer 2 Report

Comments and Suggestions for Authors

Dear Authors,

Thanks for your efforts. The recent improvements are somehow satisfactory to me. However, I have noticed several writing errors in the abstract and other sections that must be proofread again. For instance, ... protocols proposed (the "proposed protocols" is the correct), and the chosen verb has to be plural for ("excel," not "excels"). Another example of writing errors, for instance, in the first three sentences of the Introduction, you repeated "IoT" as the first subject, which is somehow wired. You can apply linking words and use another subject to make them more coherent regarding meaning connections.

Comments on the Quality of English Language

Overal, this is a well-written article, but it requires careful proofreading in terms of coherency and grammatical errors.

Author Response

Comment 1:Thanks for your efforts. The recent improvements are somehow satisfactory to me. However, I have noticed several writing errors in the abstract and other sections that must be proofread again. For instance, ... protocols proposed (the "proposed protocols" is the correct), and the chosen verb has to be plural for ("excel," not "excels"). Another example of writing errors, for instance, in the first three sentences of the Introduction, you repeated "IoT" as the first subject, which is somehow wired. You can apply linking words and use another subject to make them more coherent regarding meaning connections.

Author Response:Thank you for your careful and valuable comments and feedback. We carefully proofread the article and corrected relevant misrepresentations in the revised version.
We appreciate your careful review and believe that this revision improves the overall structure and presentation of the security analysis.
